# Research on Johnson–Cook Constitutive Model of γ-TiAl Alloy with Improved Parameters

**DOI:** 10.3390/ma16206715

**Published:** 2023-10-16

**Authors:** Limin Shi, Tong Wang, Liang Wang, Erliang Liu

**Affiliations:** 1College of Mechanical and Power Engineering, Harbin University of Science and Technology, Harbin 150080, China; 2School of Mechanical Engineering and Rail Transit, Changzhou University, Changzhou 213164, China

**Keywords:** γ-TiAl alloy, Johnson–Cook constitutive model, strain rate, thermal deformation

## Abstract

Due to its excellent physical properties, γ-TiAl alloy has been widely used in thin-walled components of aerospace engines. However, issues such as low thermal conductivity, poor machinability, and high cutting temperatures often result in difficulties in ensuring the geometric accuracy and surface integrity of the parts. This paper focuses on the study of the thermal deformation behavior of γ-TiAl alloy within a range of higher temperatures and strain rates. Firstly, by conducting quasi-static tests and Hopkinson bar tests on γ-TiAl alloy, the true stress–strain curves of γ-TiAl alloy are obtained within a temperature range of 20~500 °C and a strain rate range of 3000~11,000/s. Based on the Johnson–Cook model, the true stress–strain curves are fitted and analyzed with consideration of the coupling effect of strain rate, temperature, and strain. The strain rate hardening coefficient C and thermal softening exponent m are polynomialized, improving the Johnson–Cook constitutive model of γ-TiAl alloy. The improved model shows significant improvements in the correlation coefficient and absolute errors between the predicted values and experimental values, providing a better reflection of the thermal deformation behavior of γ-TiAl alloy within a range of higher temperatures and strain rates.

## 1. Introduction

The density of γ-TiAl alloy is only 50% that of nickel-based superalloys, yet it exhibits similar working temperatures ranging from 750 to 900 °C, making it a promising replacement for nickel-based superalloys in critical thin-walled components of next-generation aerospace engines [1,2,3,4]. However, during cutting operations, γ-TiAl alloy generates a significant amount of cutting heat, leading to severe tool wear. Exploring the mechanical behavior of γ-TiAl alloy is of great significance.

During the cutting process, substantial shear deformation occurs in the workpiece material, particularly during high-speed machining. A noteworthy characteristic of mechanical machining lies in the workpiece material experiencing exceptionally high temperatures and strain rates. Describing material behavior during such significant plastic deformation poses challenges when employing classical plasticity theories. In practice, there exists a highly nonlinear relationship between flow stress and strain, strain rate, and temperature during the machining process. Consequently, to circumvent costly and technically demanding experiments, finite element analysis (FEA) emerges as one of the most effective methods for investigating cutting mechanisms [5]. As a result, finite element simulations are commonly utilized to optimize process parameters and predict cutting performance.

Various empirical and statistical models have been proposed by previous researchers, among which the Johnson–Cook (J–C) model, developed by Johnson and Cook [6], has found widespread application in the manufacturing field. Additionally, the Power-law model [7], Zerilli–Armstrong (Z–A) model [8,9,10], and Mechanical Threshold Stress (MTS) model [9] have also been developed. Out of these models, the J–C model occupies a prominent position as one of the most extensively utilized mechanical machining plastic constitutive models. Its popularity is attributed to its favorable modeling results, simplicity, and comprehensive consideration of the effects of strain, strain rate, and temperature [11,12,13,14,15,16]. The J–C model has been successfully used by many researchers to predict machining force, temperature, and residual stress and to predict materials’ failure behaviors under various loading and temperature conditions [17].

Lee and Lin [18] explored the high-temperature deformation behavior of Ti-6Al-4V alloy using the Hopkinson bar test. To describe the flow characteristics of the material under different strains, strain rates, and temperatures, they established the Johnson–Cook constitutive model. Calamaz et al. conducted cutting tests on Ti-6Al-4V alloy and found that with increasing strain rate, the material transitioned from strain hardening to softening. They analyzed this phenomenon in relation to the dynamic recovery recrystallization mechanism of the material, and proposed the hyperbolic tangent (TANH) modified constitutive model for the Johnson–Cook model [19]. Jinhua et al. [20] proposed a reverse method for determining the Johnson–Cook parameters. They combined the experimental data of cutting forces and chip thickness and used the particle swarm optimization (PSO) algorithm to optimize the model. Through orthogonal cutting tests, they successfully identified the Johnson–Cook model for the nickel-based super-alloy Inconel 718. Zhou et al. [21] established a correlation between the parameters of the Johnson–Cook constitutive model and the cutting force, and subsequently employed the firefly optimization algorithm to enhance the constitutive model. They further utilized the unequal division parallel-sided shear zone model for orthogonal cutting to predict the cutting force. Shen et al. [22] introduced a parametric identification approach to enhance the accuracy of finite element simulation in the cutting process. They achieved this by employing a Johnson–Cook constitutive model of the material. Their research involved the establishment of various Johnson–Cook constitutive models for TC17 titanium alloy, accomplished through the use of the split Hopkinson pressure bar (SHPB) and the parameter identification method. Ren et al. [23] proposed a modified Johnson–Cook constitutive model (MJC) for Ti-6Al-4V titanium alloy, which includes a hyperbolic tangent failure function. They optimized the model by using experimental cutting forces and the added Johnson–Cook coefficients as objectives and design variables, respectively. The verification results showed that the established constitutive model accurately predicted cutting forces in accordance with practical standards. Gao et al. [24] conducted hot tensile tests on Ti-6Al-4V alloy sheets at varying temperatures (650 °C, 700 °C, and 750 °C) and different strain rates (0.1 s^−1^, 0.01 s^−1^, 0.001 s^−1^). In their research, they introduced a novel Johnson–Cook model, wherein the softening coefficient “m” was treated as a function of strain rate, and a temperature correction function was incorporated into the model. Sedaghat et al. [25] conducted a systematic analysis of four typical constitutive models: Johnson–Cook, Khan–Huang–Liang (KHL), Zerilli–Armstrong (Z–A), and Gao–Zhang (G–Z). They developed an improved constitutive model that considered the dislocation drag mechanism. The enhanced model accurately predicted the sudden increase in flow stress for all face-centered cubic (FCC), hexagonal close-packed (HCP), and body-centered cubic (BCC) materials. Özel and Karpat [26] combined the Johnson–Cook constitutive model with the cooperative particle swarm optimization (CPSO) method to study the influence of high strain rates, thermal softening, and strain rate–temperature coupling on material flow stress. They determined the constitutive model parameters using evolutionary algorithms. Four methods were employed to ascertain the measurement of flow stress data during the machining process. The initial approach involved high-speed compression testing. However, this method encountered limitations, including slow heating rates and insufficient deformation rates to prevent sample soft annealing or age hardening [27]. Furthermore, the maximum strain rate attainable was confined to approximately 10^2^ s^−1^. In response to these challenges, a separate Hopkinson pressure bar (SHPB) test was conducted to measure flow stress under higher strain rates and elevated temperature conditions [28,29]. Although this test successfully circumvented soft annealing and age hardening phenomena, the strain rates achieved remained below 10^4^ s^−1^, significantly lower than the strain rates experienced in the mechanical machining process, which typically range from approximately 10^4^ s^−1^ to 10^6^ s^−1^ [30]. Meanwhile, Xu Runrun and Jian-bo Li [31,32] separately described the thermodynamic properties of different compositions of TiAl alloys under low strain rates by constructing an Arrhenius constitutive model [31,32]. However, the machining of γ-TiAl alloy involves deformation under high temperatures and strain rates.

To investigate the thermal deformation behavior of γ-TiAl alloy during machining, this study performed quasi-static compression tests and Hopkinson pressure bar tests under high temperatures and strain rates. The Johnson–Cook constitutive model was established, and the stress–strain results were analyzed based on correlation and absolute error criteria. Furthermore, considering the coupling effect of strain rate, temperature, and strain, the constitutive model was further modified to better describe the material’s deformation process.

## 2. Materials and Methods

The chemical composition of the γ-TiAl alloy (made in Northwest Institute For Non-ferrous Metal Research, Xi’an, China) used in the experiments is displayed in Table 1. The experiments were divided into two parts: quasi-static compression tests and Hopkinson pressure bar tests. Cylindrical specimens were used for both tests, with requirements for geometric dimensions, parallelism, and perpendicularity within approximately 0.01 mm and a surface roughness *R*a of 1.6 mm.

Quasi-static compression tests at room temperature (20 °C) were conducted using a CSS electronic universal testing machine. The specimens had a diameter of ⌀5 mm and a height of 5 mm (⌀5 mm × 5 mm). The compression rate was set at 0.3 mm/min, resulting in a strain rate of 0.001 s^−1^ [33,34]. To minimize errors, each test was repeated three times.

Dynamic mechanical property tests were performed using split-Hopkinson pressure bar (SHPB) apparatus. The Hopkinson pressure bar test is bifurcated into two segments: ambient temperature impact and thermal impact. For ambient temperature impact, the Hopkinson pressure bar test is conducted at room temperature (20 °C). Subsequently, these signals are processed using the Data Proc software V1.0 to derive true stress–strain curves at different strain rates. During the test, precise control of the desired strain rates is achieved by adjusting the gas pressure within the Hopkinson pressure bar device. Conversely, for thermal impact, the specimen undergoes preheating prior to conducting the Hopkinson pressure bar test. The built-in furnace of the device is employed to heat the specimen, and once the preset temperature is reached, the Hopkinson pressure bar test is carried out. Signal collection and processing for thermal impact closely resemble those for ambient temperature impact.

The tests were conducted at temperatures ranging from 20 to 500 °C, with strain rates ranging from 3000 to 11,000 s^−1^. Three different specimen sizes, ⌀5 mm × 5 mm, ⌀4 mm × 4 mm, and ⌀2 mm × 2 mm, were used for tests at different strain rates. The ⌀5 mm × 5 mm, ⌀4 × 4 mm, and ⌀2 mm × 2 mm specimens were tested at strain rates of 3000 s^−1^, 5000 s^−1^, and values ranging from 8000 to 11,000 s^−1^, respectively. The experimental plan is described in Table 2; to reduce errors, each test group was repeated three times.

## 3. Results

The true stress–strain curve obtained from the quasi-static compression tests of the γ-TiAl alloy is shown in Figure 1. This curve reflects the axial compressive capability of the γ-TiAl alloy under quasi-static conditions and the variation of flow stress during different deformation stages. As shown, the material’s yield stage is not prominent. The stress in the plastic deformation stage increases, indicating significant strain hardening. This is because the compression process deforms and damages the lamellar structure of the material, causing distortion at the grain boundaries and hindering further slip deformation within the material. As the strain increases, the deformation resistance of the material increases.

The true stress–strain curves of the γ-TiAl alloy at temperatures of 20 °C, 200 °C, 300 °C, 400 °C, and 500 °C under different strain rates are shown in Figure 2. These curves were obtained via Hopkinson pressure bar tests. As shown, during the initial stage of deformation (elastic stage), the flow stress rapidly increases with increasing strain, exhibiting work-hardening characteristics. After reaching the peak and entering the plastic stage, the flow stress continues to increase with strain but tends to stabilize and exhibits a mostly linear trend. The rapid increase in flow stress is primarily attributed to the dominant influence of strain rate strengthening, a phenomenon commonly observed in metal materials [35,36]. From Figure 2a–e, it can be seen that at all temperature ranges, both the yield stress and flow stress increase with increasing strain rate. At the same deformation temperature, the stress increases with increasing strain rate, indicating that the γ-TiAl alloy has positive strain rate sensitivity under this test condition. Under the same temperature but different strain rates, the strain hardening rate (∂σ/∂ε) of the material is generally similar. This trend becomes more pronounced with increasing temperature.

Figure 2 shows that the initial strain value for entering the steady-state flow stage of the true stress–strain curve of the γ-TiAl alloy under different strain rate conditions varies with the test conditions. However, most curves enter the steady-state flow stage after the true strain exceeds 0.04. To study the sensitivity of γ-TiAl alloy to temperature, this study selects a segment within the true strain range of 0.04 to 0.10 from each curve as the data source for constructing the constitutive equations.

Figure 3 shows the true stress–strain curves of the material at different temperatures under strain rates of 3000 s^−1^, 5000 s^−1^, 8000 s^−1^, and 11,000 s^−1^. By comparing Figure 3a–d, it can be observed that increasing temperature leads to a decrease in yield stress and flow stress. At the same strain rate, the true stress–strain curve of the material shifts downward with increasing deformation temperature, indicating a gradual reduction in flow stress. At high strain rates, the γ-TiAl alloy exhibits significant temperature sensitivity, with plastic flow stress and yield stress gradually decreasing with increasing temperature, showing a pronounced thermal softening effect.

### 3.1. Construction of Constitutive Models

#### 3.1.1. Original Johnson–Cook Constitutive Model

The Johnson–Cook constitutive model describes the effects of deformation, strain rate, and temperature on flow stress using a multiplicative relationship. It can be fitted using experimental data and is widely used in various cutting processes and simulations owing to its intuitive simplicity and broad applicability. The Johnson–Cook constitutive model is given by
(1)$$\sigma =\left[A+B\varepsilon^{n}\right]\left[1+C\ln\left(\frac{\dot{\varepsilon }}{{\dot{\varepsilon }}_{0}}\right)\right]\left[1-T^{*}{}^{m}\right]$$
where σ represents the flow stress; ε denotes the equivalent plastic strain of the material; $\dot{\varepsilon }/{\dot{\varepsilon }}_{0}={\dot{\varepsilon }}^{*}$ is the relative equivalent plastic strain rate; ${\dot{\varepsilon }}_{0}$ is the reference strain rate chosen for quasi-static compression (here, ${\dot{\varepsilon }}_{0}$ = 0.001 s^−1^); *n* is the strain rate hardening coefficient; *C* is the logarithmic strain rate sensitivity coefficient; m is the high-temperature thermal softening exponent; and *T** = (*T* − *Tr*)/(*Tm* − *Tr*) is a dimensionless term, with *Tr* being the reference temperature (here, *Tr* = 20 °C) and *Tm* being the melting point temperature of the material. The Johnson–Cook constitutive model consists of three parts reflecting the strain hardening, strain rate hardening, and thermal softening effects of the material.

When solving the parameters in the first term of the Johnson–Cook constitutive model, quasi-static compression data are used without considering the effects of strain rate hardening and thermal softening. The stress–strain relationship at this stage can be represented as
(2)$$\sigma =A+B\varepsilon^{n}$$
where *A* is the yield stress at the reference temperature (*T_r_* = 20 °C) and reference strain rate (=0.001 s^−1^). This value can be obtained by extracting the data from the stress–strain curve obtained from quasi-static compression tests. Then, the data can be extracted from the strain hardening segment, transforming (2) to (3). By fitting the data from this segment, the values of *n* and *B* can be determined:(3)ln(σ−A)=lnB+nlnε

When solving for parameter *C*, the Hopkinson pressure bar test data at different strain rates and at room temperature (20 °C) are used without considering the thermal softening effect. At this point, the first term of the Johnson–Cook constitutive model is known, and ε represents the strain generated under the corresponding strain rate condition during the test. Because data obtained at room temperature are used, the softening term becomes 1, and (1) can be rewritten as
(4)$$\sigma =\left(A+B\varepsilon^{n}\right)\left(1+C\;\dot{\ln\varepsilon }{}^{*}\right)$$

By dividing both sides of (4) by $\left(A+B\varepsilon^{n}\right)$, we obtain
(5)$$\frac{\sigma }{\left(A+B\varepsilon^{n}\right)}=1+C\;\dot{\ln\varepsilon }{}^{*}$$

Equation (5) can be viewed as a straight line with a slope of *C* and an intercept of 1. The value of *C* is determined through data fitting. ${\dot{\varepsilon }}^{*}$ is the relative equivalent plastic strain rate.

Through the conduction of pressure bar tests at different temperatures under a specific strain rate, the parameter “*m*” can be ascertained. As the third term predominantly reflects the material’s softening effect, this approach entails treating the first two terms as known values and subsequently solving for the third term. By employing this methodology, Equation (1) is linearized, transforming into a straight line when plotted on a double logarithmic coordinate system.
(6)$$\ln\left[1-\frac{\sigma }{\left(A+B\varepsilon^{n}\right)\left(1+C\;\dot{\ln\varepsilon }{}^{*}\right)}\right]=m\ln T^{*}$$

In view of the adiabatic temperature rise phenomenon that occurs during the plastic deformation of metals at high strains and strain rates, resulting in thermal softening, it assumes a critical role in the solution for “*m*”. Consequently, it becomes imperative to ascertain the adiabatic temperature rise increment, Δ*T*, which is subsequently added to the experimental heating temperature to acquire the actual temperature.

During plastic deformation, a significant portion of the work is converted into heat, with this transformation relationship described by Equation (7):(7)kW=Q
where *W* represents the work carried out by the material during plastic deformation under impact, *Q* denotes the heat converted, and *k* is the conversion coefficient (here, *k* = 0.9). Through a series of transformations, the temperature increment Δ*T* can be determined using the expression provided:(8)$$\Delta T=\frac{k}{\rho C}\int_{0}^{\varepsilon }\sigma d\varepsilon$$

In Equation (8), *ρ* symbolizes the material density, *C* denotes the material’s specific heat capacity, *σ* represents the true stress, and *ε* signifies the true strain. When considering the temperature increment Δ*T* during the plastic deformation process, the actual temperature of the material can be expressed as shown in Equation (9):(9)$$T=T_{0}+\Delta T$$

The melting point of γ-TiAl alloy is approximately 1460 °C, which is used as the reference for calculating the melting point. Considering the effect of the adiabatic temperature rise, the actual temperature can be fitted to determine the value of *m*.

Using the aforementioned method, the coefficients of the Johnson–Cook constitutive model for γ-TiAl alloy are calculated as follows: *A* = 927, *B* = 1394.5, *n* = 1.1816, *C* = 0.0318, *m* = 1.6635. The resulting constitutive model is represented by (10):(10)$$\sigma =\left(927+1394.5\varepsilon^{1.1816}\right)\left(1+0.0318\;\dot{\ln\varepsilon }{}^{*}\right)\left(1-T^{*1.6635}\right)$$

#### 3.1.2. Analysis of the Fitting Results of the Original Johnson–Cook Constitutive Model

Based on (10), comparison graphs of the calculated flow stress values from the original Johnson–Cook constitutive model and the experimental values are shown in Figure 4. The closer the predicted values are to the experimental values, the better the agreement. Therefore, it is necessary to analyze and explain the conformity of the predicted results and the experimental results.

Correlation coefficient (or correlation) is a commonly used statistical indicator that represents the degree of agreement between calculated values and experimental values. Its mathematical expression is as follows:(11)$$R=\frac{\sum_{i=1}^{n}\left(x_{i}-\bar{x}\right)\left(y_{i}-\bar{y}\right)}{\sqrt{\sum_{i=1}^{n}{\left(x_{i}-\bar{x}\right)}^{2}•\sum_{i=1}^{n}{\left(y_{i}-\bar{y}\right)}^{2}}}=\frac{n\sum_{i=1}^{n}x_{i}y_{i}-\sum_{i=1}^{n}x_{i}•\sum_{i=1}^{n}y_{i}}{\sqrt{n\sum_{i=1}^{n}x_{i}{}^{2}-{\left(\sum_{i=1}^{n}x_{i}\right)}^{2}}•\sqrt{n\sum_{i=1}^{n}y_{i}{}^{2}-{\left(\sum_{i=1}^{n}y_{i}\right)}^{2}}}$$
where *n* is the total number of observation points; *x_i_* is the experimentally measured flow stress; *y_i_* is the predicted value of the model; and *x* and *y* are the mean values of *x_i_* and *y_i_*, respectively. However, using the correlation coefficient alone cannot fully assess the goodness of fit of the calculation results. It is also necessary to introduce the absolute error to indicate the deviation between the calculation results and the experimental values. The absolute error can be expressed as
(12)$$\Delta =\frac{1}{n}\sum_{i=1}^{n}\left|x_{i}-y_{i}\right|\times 100\%$$

The results of the correlation coefficient and absolute error for the calculated original constitutive model are shown in Table 3.

Figure 4 compares the calculated flow stress values of the original Johnson–Cook constitutive model with the experimental values. Figure 4a–d represent the comparison curves under different strain rates: 3000 s^−1^, 5000 s^−1^, 8000 s^−1^, and 11,000 s^−1^, respectively. These figures show that the curve errors are small at strain rates of 5000 s^−1^ and 8000 s^−1^, whereas they are larger at strain rates of 3000 s^−1^ and 11,000 s^−1^. Additionally, as the strain rate increases, the calculated results of the original constitutive model gradually become lower than the experimental values, and the calculated curve gradually becomes less steep than the experimental curve.

Figure 5 represents the dispersion of stress calculated using the original Johnson–Cook constitutive model compared to the experimental stress, that is, the correlation between the two sets of values. The red line in the figure represents a correlation coefficient of 1 (100%). The points above the line represents perfect agreement between the simulated and experimental values, while the points below the line indicate that the simulated values are lower than the experimental values. This figure includes data under different strain rate conditions. The calculated values of the flow stress for the γ-TiAl alloy using the original Johnson–Cook constitutive model are both higher and lower than the experimental values, with significant deviations. The correlation coefficient is only 0.838.

Because of the underlying assumptions that it is built upon, the original Johnson–Cook constitutive model cannot accurately describe the rheological behavior of γ-TiAl alloy during the cutting process. These assumptions assume that strain hardening, strain hardening rate, and heat-softening effects are independent and do not interfere with each other. In reality, the strain hardening coefficient C of γ-TiAl alloy varies with temperature and strain. To describe the rheological behavior of γ-TiAl alloy more accurately, the relationship between these three factors and their interaction with flow stress must be considered. This was carried out to improve the Johnson–Cook constitutive model and accurately describe the rheological behavior of the γ-TiAl alloy during the cutting process.

#### 3.1.3. Improvement of the Original Johnson–Cook Constitutive Model

To further improve the predictive ability of the model, the original Johnson–Cook constitutive model was modified. According to (1), the first term, $\left[A+B\varepsilon^{n}\right]$, dominates the distribution range of stress and determines the approximate stress distribution. The last two terms, $\left[1+C\ln\left(\frac{\dot{\varepsilon }}{{\dot{\varepsilon }}_{0}}\right)\right]$ and $\left[1-T^{*}{}^{m}\right]$, have a value of 1 at the reference temperature and reference strain. When the temperature is the reference temperature and the strain rate is the reference strain, these two terms have no influence on the calculation results. However, when the temperature and strain rate deviate from the reference point, these two terms influence the calculated flow stress. The strain rate hardening coefficient *C* and the thermal softening exponent *m* determine the magnitude of the adjustment. In the original model, parameter *C* was introduced as a constant, but *C* is actually not a constant; it varies with temperature and strain rate. The second term must consider the coupling effect of strain rate, temperature, and strain.

The improved Johnson–Cook constitutive model is formulated as
(13)$$\sigma =\left[A+B\varepsilon^{n}\right]\left[1+C_{1}\ln\left(\frac{\dot{\varepsilon }}{{\dot{\varepsilon }}_{0}}\right)+C_{2}\ln{\left(\frac{\dot{\varepsilon }}{{\dot{\varepsilon }}_{0}}\right)}^{2}\right]\left[1-T^{*}{}^{M}\right]$$
where the second term is used to describe the material’s strain rate hardening, considering the coupling effect of strain rate, temperature, and strain. Parameters *C*_1_ and *C*_2_ vary with strain. The third term describes the influence of temperature on material softening, and parameter *M* is an undetermined coefficient. The specific forms of parameters *C*_1_, *C*_2_, and *M* are quadratic polynomials:(14)$$C_{1}=c_{0}+c_{1}\varepsilon +c_{2}\varepsilon^{2}$$
(15)$$C_{2}=c_{3}+c_{4}\varepsilon +c_{5}\varepsilon^{2}$$
(16)$$M=m_{0}+m_{1}\ln\left(\frac{\dot{\varepsilon }}{{\dot{\varepsilon }}_{0}}\right)+m_{2}\ln{\left(\frac{\dot{\varepsilon }}{{\dot{\varepsilon }}_{0}}\right)}^{2}$$
where *c*_0_–*c*_5_ and *m*_0_–*m*_2_ are undetermined coefficients.

The parameter values of the modified first term are the same as those of the original Johnson–Cook constitutive model. When solving the second term, SHPB test data at room temperature are used without considering the effect of temperature softening. In this case, the value of the third term is 1, and (13) becomes (17):(17)$$\sigma =\left[A+B\varepsilon^{n}\right]\left[1+C_{1}\ln\left(\frac{\dot{\varepsilon }}{{\dot{\varepsilon }}_{0}}\right)+C_{2}\ln{\left(\frac{\dot{\varepsilon }}{{\dot{\varepsilon }}_{0}}\right)}^{2}\right]$$

By transforming (17) into (18), different values of *C*_1_ and *C*_2_ are obtained for different strains. These values are then subjected to quadratic fitting to obtain *c*_0_–*c*_5_, as shown in Table 4.
(18)$$\frac{\sigma }{A+B\varepsilon^{n}}=1+C_{1}\ln\left(\frac{\dot{\varepsilon }}{{\dot{\varepsilon }}_{0}}\right)+C_{2}\ln{\left(\frac{\dot{\varepsilon }}{{\dot{\varepsilon }}_{0}}\right)}^{2}$$

When solving for parameter *M*, experimental results at various strain rates and temperatures are used. The method described in (6) is used to solve for values of *M* at different strain rates, and then, *M* is subjected to binomial fitting to obtain *m*_0_–*m*_2_, as shown in Table 5.

Figure 6 compares the calculated flow stress values of the improved Johnson–Cook constitutive model with the experimental values. Compared to the original model, the improved model shows a high level of overall fit between the calculated values and experimental values for the flow stress of the material at four different strain rates: 3000 s^−1^, 5000 s^−1^, 8000 s^−1^, and 11,000 s^−1^.

Figure 7 compares the average curve of experimental values with the fitted average curve of the improved Johnson–Cook constitutive model. It can be clearly seen that the majority of the points are close to a straight line. Compared to the original Johnson–Cook constitutive model, the improved model accurately calculates the true stress values. This is because the polynomial function for the strain rate hardening coefficient C, which considers the coupling effect of strain rate, temperature, and strain, provides a more accurate calculation of the strain rate hardening coefficient.

Table 6 shows that the improved JC model has a significantly improved correlation coefficient and absolute error compared to the original model. The correlation coefficient of the improved model is 0.994, higher than the original model, and the absolute error is 11.57, lower than the original model. This indicates that the improved model can effectively represent the high-temperature thermal deformation behavior of γ-TiAl alloy during high-speed cutting.

## 4. Conclusions

This study conducted quasi-static compression tests at room temperature and SHPB tests at high temperatures and high strain rates on γ-TiAl alloy. The following conclusions were drawn from the analysis:(1)By conducting mechanical property tests on the γ-TiAl alloy, encompassing a temperature range from 20 to 500 °C and strain rates ranging from 0.001 to 11,000 s^−1^, we successfully obtained the true stress–strain relationships of the material at both room temperature and high temperature under high strain rates. The test results indicate that in the initial stage of deformation (elastic stage), the flow stress increased rapidly with increasing strain. Upon reaching the peak and entering the plastic stage, the flow stress continued to rise with strain but eventually approached a state of stabilization, exhibiting an almost linear trend. Additionally, at a constant deformation temperature, the stress exhibited an increase with higher strain rates. Conversely, under identical strain rates but varying temperatures, elevated temperatures resulted in a reduction in true stress.(2)In order to acquire the Johnson–Cook constitutive model parameters for γ-TiAl alloy, an initial Johnson–Cook constitutive model was formulated, accounting for the coupling effect of strain, temperature, and strain rate under high strain rates and within different temperature ranges. Subsequently, this original Johnson–Cook model underwent refinement, leading to the derivation of pertinent parameters and resulting in a noteworthy enhancement of the model’s correlation coefficient from 0.838 to 0.994, along with a reduction in the absolute error from 61.49 to 11.57. This substantial improvement significantly enhanced the model’s capability to accurately describe the thermal deformation behavior of γ-TiAl alloy over a wide range of temperatures and strain rates.

## Figures and Tables

**Figure 1 materials-16-06715-f001:**
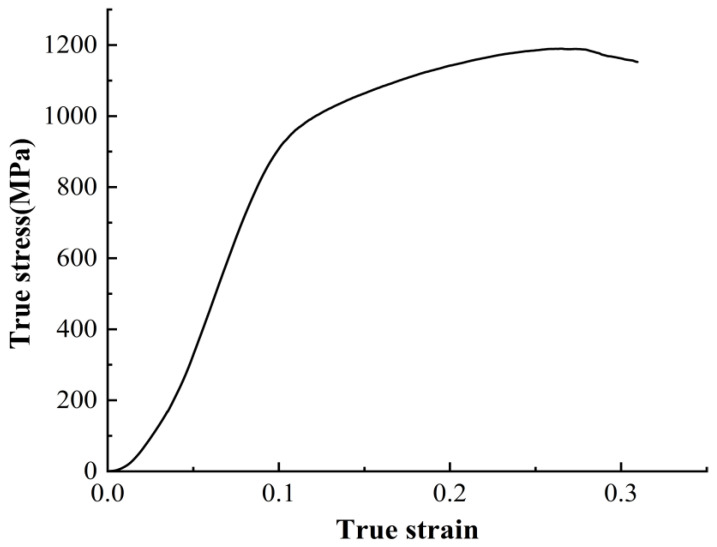
True stress–strain curve of γ-TiAl alloy under quasi-static conditions.

**Figure 2 materials-16-06715-f002:**
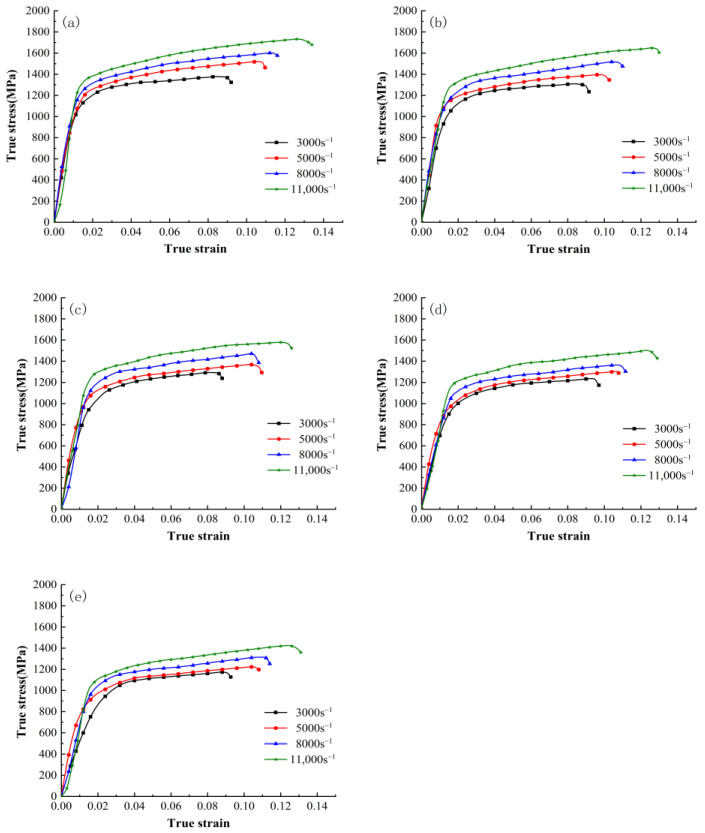
True stress–strain curves of γ-TiAl alloy under different temperatures: (**a**) 20 °C; (**b**) 200 °C; (**c**) 300 °C; (**d**) 400 °C; (**e**) 500 °C.

**Figure 3 materials-16-06715-f003:**
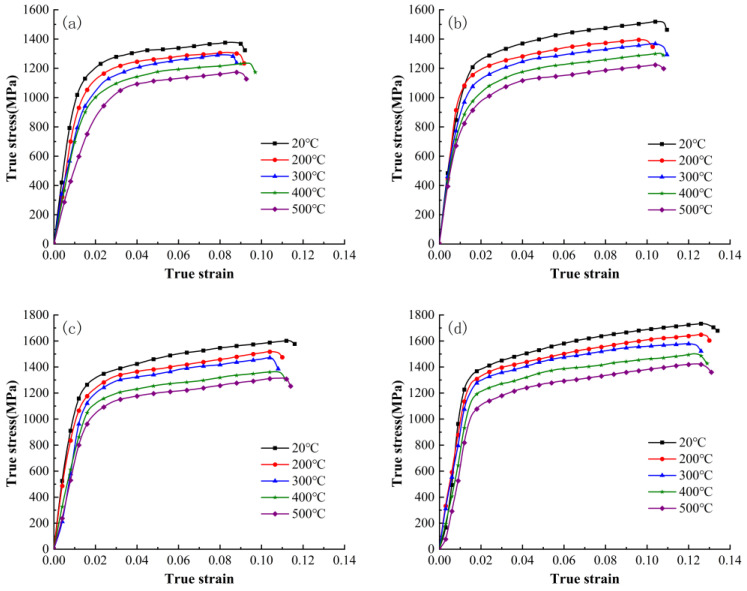
True stress–strain curves of γ-TiAl alloy under different strain rates: (**a**) 3000 s^−1^; (**b**) 5000 s^−1^; (**c**) 8000 s^−1^; (**d**) 11,000 s^−1^.

**Figure 4 materials-16-06715-f004:**
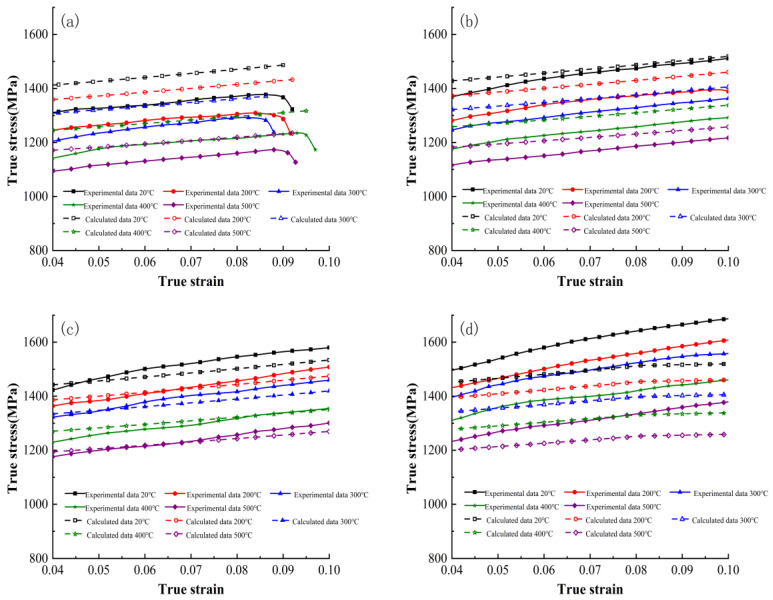
Comparison between calculated flow stress values from the original Johnson–Cook constitutive model and the experimental values. (**a**) 3000 s^−1^; (**b**) 5000 s^−1^; (**c**) 8000 s^−1^; (**d**) 11,000 s^−1^.

**Figure 5 materials-16-06715-f005:**
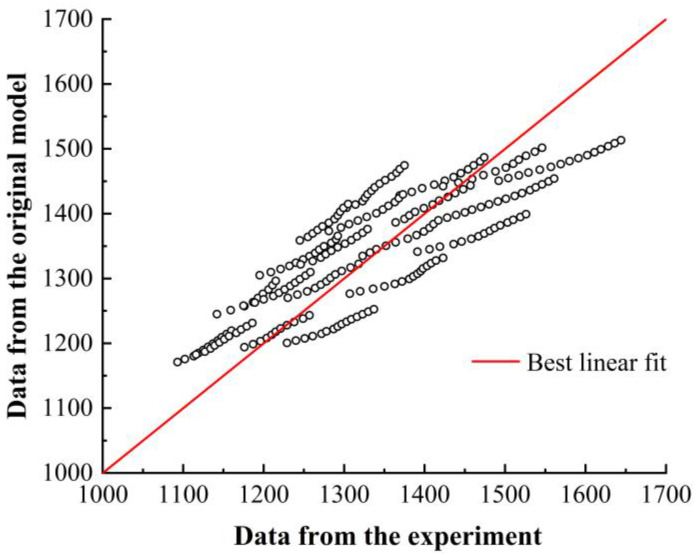
Correlation between predicted values and experimental values of the original Johnson–Cook constitutive model.

**Figure 6 materials-16-06715-f006:**
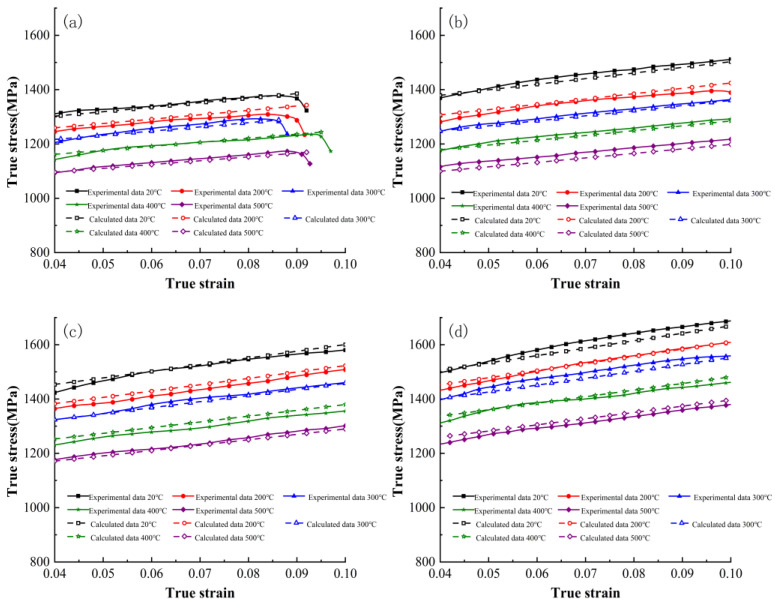
Comparison of calculated flow stress values of the improved Johnson–Cook constitutive model with experimental values: (**a**) 3000 s^−1^; (**b**) 5000 s^−1^; (**c**) 8000 s^−1^; (**d**) 11,000 s^−1^.

**Figure 7 materials-16-06715-f007:**
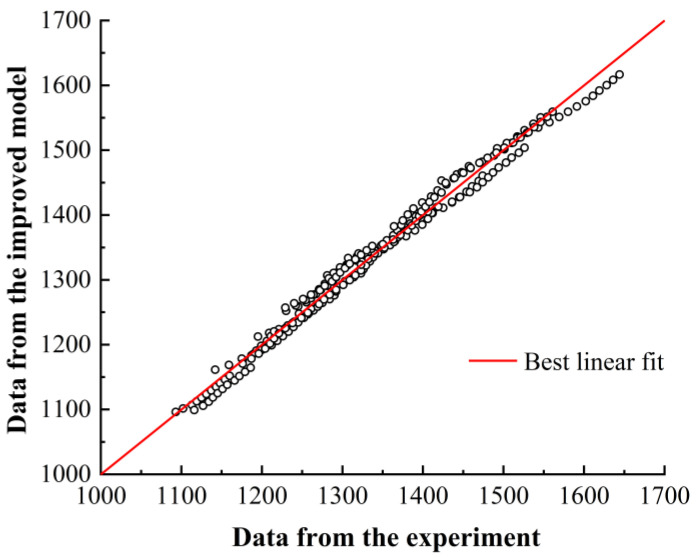
Comparison of predicted values and experimental values using the improved constitutive model.

**Table 1 materials-16-06715-t001:** Chemical composition (mass fraction, %) of γ-TiAl alloy.

Al	Cr	Nb	C	H	N	O	Ti
32.6	2.55	4.69	<0.004	0.0007	0.004	0.054	The remainder

**Table 2 materials-16-06715-t002:** Hopkinson pressure bar test plan.

Specimen Size (mm)	Strain Rate (s^−1^)	Temperature (°C)
⌀5 mm × 5 mm	3000	20, 200, 300, 400, 500
⌀4 mm × 4 mm	5000	20, 200, 300, 400, 500
⌀2 mm × 2 mm	8000, 11,000	20, 200, 300, 400, 500

**Table 3 materials-16-06715-t003:** Correlation coefficient and absolute error of the original models.

Constitutive Model	Correlation	Absolute Error
Original	0.838	61.49

**Table 4 materials-16-06715-t004:** Coefficients of parameters *C*_1_ and *C*_2_ in the improved Johnson–Cook constitutive model.

*c* _0_	*c* _1_	*c* _2_	*c* _3_	*c* _4_	*c* _5_
−0.08003	−0.62607	0.54589	0.00692	0.04419	−0.04329

**Table 5 materials-16-06715-t005:** Coefficients of parameter *M* in the improved Johnson–Cook constitutive model.

*m* _0_	*m* _1_	*m* _2_
133.96	−17.0043	0.5457

**Table 6 materials-16-06715-t006:** Correlation and absolute error of the two constitutive models.

Constitutive Model	Correlation	Absolute Error
Original	0.838	61.49
Improved	0.994	11.57

## Data Availability

Not applicable.

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
