# Peer review of "Research on Johnson–Cook Constitutive Model of γ-TiAl Alloy with Improved Parameters"

_materials, 2023, doi:10.3390/ma16206715_

Round 1

Reviewer 1 Report

In this study, experimental tests were conducted to analyze the deformation behavior of γ-TiAl alloy under elevated temperatures and high strain rates. Furthermore, a modified Johnson-Cook model was utilized to simulate the behavior, taking into account the coupling effect of strain rate, temperature, and strain. The topic is interesting and would be of interest to the readers of the Journal of Materials. The findings are also good, and the paper makes an acceptable contribution. Hence, the manuscript is recommended for publication with minor revisions. The few comments that need to be addressed are provided below:

1- Could you elaborate on the factors contributing to the observed stabilization of flow stress in the plastic deformation phase?

2-Are there any limitations or conditions under which the model may have reduced accuracy?

3-Did the study investigate any potential mechanisms behind the coupling effects?

4-Could you discuss potential future research directions or applications that could benefit from this enhanced model's predictive capabilities?

Reviewer 2 Report

The manuscript deals with a development of a modified constitutive model for a TiAl alloy.

The abstract contains too long uncomprehensive sentences and should be modified. The introduction is a little too long, too. The information provided is OK, however, identical problem as in the abstract arises - the sentences are somewhat long and clumsy… Please reconsider the stylistics throughout the manuscript.

Again, in the Conclusions – this section should contain the bullet points arising from the conducted research, rather than a lengthy summary of the works done, and, by the way, the results acquired. The majority of this section is discussion – the authors could consider creating a separate section of discussion, into which the information can be added.

In summary, the provided data is all right, but the form of presenting is clumsy and the stylistics should be improved.

Stylistics should be significantly improved.

Reviewer 3 Report

Title: "Research on Johnson-Cook Constitutive Model of γ-TiAl Alloy with Improved Parameters"

The paper titled "Research on Johnson-Cook Constitutive Model of γ-TiAl Alloy with Improved Parameters" investigates the high-temperature deformation behavior of γ-TiAl alloy across a range of elevated temperatures and high strain rates. Overall, the paper presents valuable insights; however, several aspects require further clarification and analysis.

 Specimen Size Selection: The paper mentions the use of different specimen sizes (⌀5×5 mm, ⌀4×4 mm, ⌀2×2 mm) for tests at various strain rates. It would be beneficial to elucidate the criteria employed in choosing these sizes and whether the choice had a significant impact on the results. This information is crucial for the repeatability and applicability of the findings.

 Adiabatic Temperature Rise: The mention of adiabatic temperature rise during plastic deformation raises questions about the mechanisms considered for heat dissipation during experiments. A comprehensive explanation of how adiabatic temperature rise was managed, controlled, or mitigated is essential to understand its influence on the results.

 Thermal Softening: The paper discusses thermal softening at high temperatures. To enhance the paper's clarity, it would be valuable to identify any critical temperature thresholds at which this effect becomes particularly pronounced. Additionally, explaining how this thermal softening phenomenon was addressed within the developed constitutive model will provide deeper insights into its practical implications.

 Statistical Analysis of Repetition: While the paper mentions that each test group was repeated three times to reduce errors, it would be beneficial to delve into the specific statistical analyses conducted to confirm the consistency of the results between repetitions. This information can demonstrate the reliability and robustness of the experimental data.

 Microstructure Influence: Consideration of the material's microstructure, such as grain size and distribution, on its deformation behavior at high temperatures and strain rates is pivotal. The paper should provide insights into whether the study accounted for these microstructural factors and, if so, how they were factored into the analysis.

 Alloy Composition and Impurities: Variations in alloy composition or the presence of impurities within the γ-TiAl alloy can significantly impact deformation behavior. The review should clarify whether these factors were controlled, considered, or quantified in the analysis and highlight their potential effects on the results.

 In conclusion, the paper offers valuable contributions to our understanding of γ-TiAl alloy's high-temperature deformation behavior. Addressing the questions and suggestions outlined above will not only enhance the paper's clarity but also contribute to its scientific rigor and applicability in practical aerospace and materials engineering contexts.

Round 2

Reviewer 2 Report

The authors have addressed all the comments.

English has been improved, however, final proofreading is suggested.

Reviewer 3 Report

The requested modifications have been made correctly. Congratulations to all the authors.